# Time-Varying Wear Calculation Method for Fractal Rough Surfaces of Friction Pairs

Qiang Hao, Jian Yin, Yu Liu, Lu Jin, Shengfang Zhang and Zhihua Sha *

College of Mechanical Engineering, Dalian Jiaotong University, Dalian 116028, China
* Correspondence: zhsha@djtu.edu.cn

**Abstract:** For the wear problem of the real rough surface during sliding friction, based on fractal theory and Hertz contact theory, a 3-D fractal rough surface with random characteristics is constructed, and the relationship between the wear deformation depth of the rough peak and its real contact area during the wear process is derived. Furthermore, considering the peak wear and pit scratch phenomena of rough surfaces in different contact states, the time-varying wear calculation model of the worn surface and the compensation wear calculation model of the unworn surface are established, respectively, and the relationship between the instantaneous wear amount and the dynamic change in the rough surface topography is comprehensively characterized. Combined with image digitization technology, the 3-D rough surface is converted into a 2-D discrete plane with 3-D information. According to the dynamic real-time update of the graph data, the iterative calculation of the wear cycle is completed, the time-varying wear calculation method for fractal rough surfaces of friction pairs is proposed, and the dynamic change in the wear amount and surface topography of the rough surface is simulated. The simulation results are experimentally verified and the influence of friction parameters on the surface topography is analyzed. The results show that after the wear simulation, the profile height of the rough surface is reduced, and the average wear depth is 0.02 mm. Increases in rotational speeds and external loads can exacerbate surface wear, surface topography tends to be flattened, and surface carrying capacity increases. This provides theoretical guidance for the development and manufacture of friction pairs.

**Keywords:** friction; wear calculation; rough surface; fractal theory; surface topography

## 1. Introduction

With the development of the economy and the improvement of science and technology, the work of transportation equipment is heavy, so it puts forward higher requirements for the safety and security of mechanical equipment operation. Disc brakes are key equipment to ensure the safe operation of transport such as rail trains and buses. When the mechanical equipment is faced with the need to execute the parking order or emergency braking conditions in the actual work, the meeting adopts the mechanical braking method [1]. Disc brakes are common brake friction pairs with tribological properties such as surface damage and material wear. This phenomenon is common in the braking process of rail trains or vehicles and other traffic transportation. The uneven stress of the friction pair caused by surface wear will lead to irregular wear, reduce the braking efficiency, and seriously affect the driving safety [2]. In particular, for brake friction pairs in long-term service, surface wear is an important indicator to measure the safe operation of disc brakes. Surface wear is the inevitable result of the mechanical friction pair of mechanical equipment in the friction process; it is also one of the key factors that lead to the failure of the mechanical friction pair; the worn surface morphological characteristics and its change law is an important basis for reacting to the working state of the friction pair. Therefore, an in-depth analysis of the surface wear mechanism of the friction pair is an important part of the study of mechanical condition prediction and fault diagnosis. Many factors affect surface wear, which makes

the wear mechanism complex. The research on the friction and wear mechanism of the friction pair surface has become the research focus of scholars around the world.

The condition of the wear phenomenon is that the two solid surfaces in contact with each other have a relative sliding state. Archard proposed a classical Archard wear calculation model considering the contact pressure, the relative speed, and the hardness of the material, and the contact state of the surfaces was represented by the wear coefficient [3,4]. However, during the real wear process, the contact surfaces are rough. The rough surface introduces the complexity of the surface topography and strongly influences the fracture and destruction of materials, which defines contact stiffness and plays a dominant role in the friction and wear process. In addition, the micro-mechanical processes that produce the small-scale surface topography are so complex. It is often reasonable to model roughness as random processes that may be stationary or non-stationary.

Ramírez et al. summarized some methods used for the simulation of randomly rough surfaces and their characterization. Their numerical efficiency allowed for large-scale statistical simulation of contact behavior, which was analyzed through a generalized version of the classical Greenwood–Williamson model [5]. Wang et al. developed a simulation method for generating non-Gaussian rough surfaces with desired autocorrelation function and spatial statistical parameters, including skewness and kurtosis. It was convenient to simulate the non-Gaussian rough surfaces with various types of autocorrelation functions and large autocorrelation lengths [6]. Aslyamov et al. proposed a new theoretical approach to obtaining the nanoscale morphology of rough surfaces. The method was based on one of the most realistic models of rough surfaces formulated in terms of random correlated processes. From this procedure, the best-fit detailed geometry of rough surfaces was obtained [7]. It can be found that the construction of the real rough surface is constantly improving, and the construction of the model surface in the way of probability and statistics makes the model surface closer to the real surface. However, the research shows that the topography of the wear surface has unstable randomness, statistical self-similarity, and self-affineness. As a result, the statistical-based roughness characterization parameters show instability with changes in test conditions. The fractal parameters overcome the deficiency in the scale correlation of traditional characterization parameters to a certain extent, and can effectively characterize the wear surface. However, for the fractal representation of the rough surface, it is necessary to determine the proper fractal dimension calculation method.

The analysis of rough surface morphology plays an important role in the functional characteristics of the contact surface of mechanical parts. Therefore, considering the deformation effect of the asperity on the rough surface during the friction process, Shen et al. studied the effect of the power spectral density method based on the Monte Carlo method. The result showed that the power spectral density method had a good characterization effect on the fractal simulation contour curve [8]. Sun et al. presented the critical deformation estimation model, which expressed critical deformation as a function of fractal parameters and contact deformation. Furthermore, the contact stiffness calculation model of single asperity was brought forward by considering critical deformation change. Owing to the possibility of plastic deformation during the loading process, the experimental curve was nonlinear, resulting in an error between the experimental results and the theoretical calculation results [9]. Liu et al. aimed to improve the computational accuracy of the fractal dimension of the profile curve, and a two-stage method based on the power spectrum method and the structure-function method was proposed [10]. Dong et al. developed a numerical model to address the stress concentrations at the roughened surface/interface under contact loading in plane-strain conditions. The model provided a useful tool to accurately capture the stresses within the roughened areas [11]. Mirsalimov and Akhundova constructed a model of a rough friction surface, so the displacement function of the external contour points of a hub and the friction surface microgeometry that correspond to the uniform distribution of the contact pressure over a friction surface were theoretically determined [12]. It can be found that the rough surface morphology and structure can indeed affect the surface friction performance of the friction pair.

Considering the influence of the contact state of the friction pair surface, the wear calculation model of the rough surface could be established. Furthermore, to develop predictive wear laws, relevant material parameters and their influence on the wear rate need to be identified. Brink et al. proposed a simple model for adhesive wear in dry sliding conditions. A novelty of the model was the explicit tracking of the sliding process, which meaningfully connected particle emission rates and sizes to the macroscopic wear rate [13]. Li et al. developed a numerical model to predict the generation of wear debris with given rough surfaces [14]. Liu et al. established the wear model based on the relationship between real-time wear rate and real-time surface roughness during the test. It was found that the tribological behaviors strongly depended on the real-time roughness [15]. Meanwhile, Springis et al. proposed a suitable wear calculation model, allowing consideration of a set of parameters necessary to calculate the sliding friction pair. The proposed model for wear calculation was based on the application of theories from several branches of science to the description of 3-D surface micro-topography [16]. Therefore, to represent the real situation more closely, the effect of rough surfaces on wear was considered. Wu et al. presented a novel 3-D characterization of the wear effects of a roll surface on the texture transfer in skin-pass rolling. The study revealed that the roughness scale of a rolled metal strip decreased gradually due to the roll surface wear, and the rolling force, relative sliding, and rolling speed at the roll–strip interface significantly affected the roll surface wear [17]. Shi et al. analyzed the effects of surface topographic parameters on friction with friction experiments. The wear characteristics of the samples were mainly characterized as strain fatigue, grinding, and scraping. The results provided a theoretical basis for the functional characterization of surface topography [18]. The research shows that the influence factors of rough surface wear are not only the friction parameters but also the changes in surface morphology.

During the friction process of rough surfaces, the surface contact state changes dynamically, which makes friction parameters such as the real contact area unstable. Modeling the real contact area plays a key role in every tribological process, such as friction, adhesion, and wear. Contact between two solids does not necessarily occur everywhere within the apparent contact area. Emami et al. proposed a different approach to correct the over-estimation of vertical displacement in Persson's contact theory for rough surfaces with self-affine fractal properties. The main advantage of the proposed method was that it used physical parameters such as the surface roughness characteristics, material properties, sliding velocity, and normal load to correct the model [19]. Izmailov found that the discrete adhesive contact model made it possible to calculate the main operational characteristics of the contact joint: deformation and the actual contact area [20]. Wei et al. proposed a fractal stratified characterization method to evaluate the surface profile evolution of a plateau honing cylinder liner during the wear process. Experimental results demonstrated the ability of the proposed method to outperform the traditional fractal method in characterizing the evolution of the plateau honing cylinder liner surface profile during the wearing process [21]. Due to the wear process, the initial anisotropic response evolves with the variation in asperity distribution, tending to a steady-state pattern. Mróz et al. presented the analysis of transient and quasi-steady states of coupled friction and wear processes for the case of surface anisotropy. The orthotropic surface roughness with a parallel layout of asperity ridges was considered [22].

Based on the research on the law of wear, the influencing factors of wear calculation have also been analyzed. Berardo and Pugno proposed an analytical model to describe the anisotropic friction, adhesion, and wear of hierarchical surfaces. The model could describe friction between two generic rough surfaces in contact, sliding one against the other [23]. Huang et al. developed and theoretically validated a new methodology for describing Gaussian rough surfaces based on the Stribeck curve. The results showed that the type of friction pair influenced greatly the wear volume and wear rate of contact surfaces [24]. Therefore, Makhkamov considered the problems of calculating the wear on the work surfaces of sliding friction units based on energy-friction theory. The derived results made

it possible to calculate the amount of wear on friction sliding surfaces [25]. Furthermore, Aghababaei and Zhao performed systematic long-timescale asperity-level wear simulations using two recently developed coarse-grained model potentials with extremely brittle and ductile behavior. A linear wear relation could be recovered from simulations only when the material removal progresses by plastic deformation at the asperity tip, confirming the long-standing theoretical hypothesis made by Archard [26]. Poll et al. transferred the global Archard equation into a numerical wear model, which allowed a spatially resolved determination of wear depth for dry and boundary lubricated contacts [27].

However, during the friction contact process of the actual friction pair, the wear phenomenon of the rough surface would lead to the change in the surface topography, which, in turn, affects the real contact area of the friction pair. Therefore, the wear area of the rough surface changes dynamically during the friction process, and it is of great significance to study the change laws of the rough surface wear and worn surface topography during friction.

Considering the effect of real-time dynamic change of rough surface topography on surface wear, this paper uses a combination of numerical simulation and experiment to study the change laws of the surface topography during the wear process. A time-varying wear calculation simulation method is proposed, which includes a time-varying wear calculation model for a worn surface and a compensation wear calculation model for an unworn surface. The numerical model of wear calculation is established. The control variable method is used to analyze the influence of the rotational speed and the external load on the characteristic parameters of surface topography, which provides a new approach for solving tribological problems.

## 2. Method of Wear Calculation for Friction Pair

Surface research is the basis for the development of tribology. For the tribological systems that have been studied, in addition to the physical and chemical properties of the surface affecting the friction and wear mechanisms, the surface morphology of the friction surface also affects the surface friction properties under the action of mechanical forces. Therefore, considering the influence of surface topography on surface wear, the tribological simulation modeling of the friction pair surface is divided into the mathematical modeling of surface topography and wear calculation of the friction process. The friction motion of the friction pair is the process of mutual extrusion and shearing of the rough peaks between the contact surfaces, so the surface topography formed by the rough peaks has a significant influence on the wear amount.

### 2.1. Friction Contacts of Rough Surfaces

The surface of the mechanical friction pair has a surface roughness that meets the actual use requirements and has uneven structural characteristics, which affect the friction, wear, and lubrication properties of the friction pair. During the contact process of the friction pair surfaces, the high peaks of the rough surface come into contact first, and the ratio of the real contact area to the nominal contact area is small. Mandelbort proposed the Weierstrass–Mandelbrot function (W–M function). Further, according to the W–M fractal function [28], the topographical structure of the rough surface has fractal characteristics, and the part and the whole-surface contours curve have self-similarity. However, the W–M fractal function is only limited to the characterization of the rough surface profile on the 2-D scale and cannot express the changes in the rough surface on the 3-D scale. Yan and Komvopoulos derived the functional model of the 3-D fractal surface based on the W–M function [29], that is:

$$\begin{cases} z(x,y) = L\left(\frac{G}{L}\right)^{D-2}\left(\frac{\ln\gamma}{M_0}\right)^{\frac{1}{2}} \cdot \sum_{m=1}^{M_0}\sum_{0}^{n_{\max}}\gamma^{(D-3)n}\left\{\cos\varphi_{m,n} - \cos\left[\frac{2\pi\gamma^n(x^2+y^2)^{\frac{1}{2}}}{L}\cos\left(\tan^{-1}\frac{y}{x} - \frac{\pi m}{M_0}\right) + \varphi_{m,n}\right]\right\} \\ n_{\max} = \mathrm{int}[\log(L/L_s)/\log\gamma] \\ D \in (2,3) \\ \varphi_{m,n} \in (0,2\pi) \end{cases} \quad (1)$$

where $z(x,y)$ is the surface contours height of the rough surface, and $x$ and $y$ are surface contours geometric coordinates. $G$ is the scaling coefficient, and $D$ is the 3-D fractal dimension. $\gamma$ is a parameter that determines the density of frequencies of the surface. $M_0$ denotes the number of superposed ridges used to construct the surface. $m$ and $n$ are frequency indices of random contours. $L$ is the sample length, and $L_s$ is the cut-off length. $n_{\max}$ is the highest frequency of the spatial frequency index. $\varphi_{m,n}$ is a uniformly distributed value of the random phase. Therefore, rough surfaces with fractal features can be established by the W–M function.

The microstructure of the rough surface is composed of peaks and pits. During the friction process, the peaks contact each other and the wear phenomenon of the surface material occurs. To facilitate the analysis of the friction mechanism of the rough surface in the friction contact process, the rough surface contact relationship of the friction pair is simplified as the contact between the rough surface and the rigid ideal plane, and the following assumptions are made: the surface strengthening effect caused by elastic contact during the contact process is ignored; the material hardness variation with the change in the surface depth is ignored; interactions between adjacent convex peaks are ignored. As shown in Figure 1, based on the surface profile method, the rough surface peak profile is identified, the unworn surface is used as the reference surface, and the difference in peaks between the unworn surface and the worn surface is used to represent the wear amount.

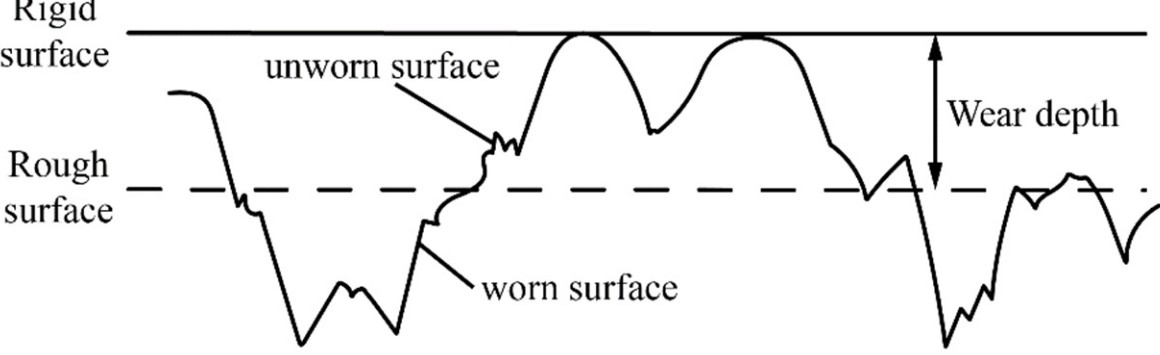

**Figure 1.** Schematic of wear characterization of rough surfaces.

At the microscopic scale, when friction occurs between two solid surfaces, the real contact of rough surfaces is concentrated in the coincident regions of peaks. Under the action of friction, the deformation and wear phenomenon of peaks occur, which changes the real contact area of the rough surface. The area size of the real contact area and the distribution of the contact location have a decisive influence on the friction and wear of the surface.

The peak shape of the rough surface is generally ellipsoid or cone. As the contact area of the peak is much smaller than its curvature radius, the peak shape of the micro-convex peak is simplified as a sphere, as shown in Figure 2.

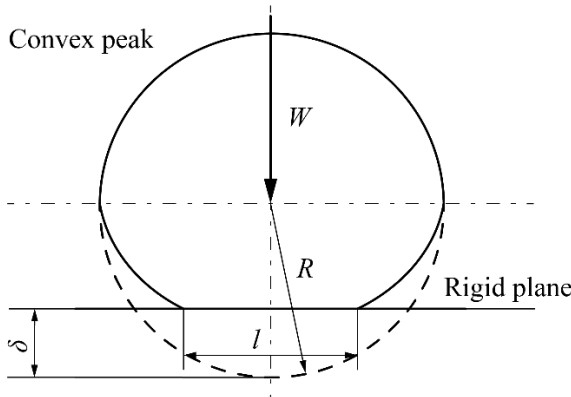

**Figure 2.** Deformation mechanism of the peak.

In the contact process between the peak and the rigid plane, the normal deformation is generated under the external load. According to the Hertz contact theory [30], that is:

$$
\begin{cases}
\delta = \sqrt[3]{\dfrac{9W^2}{16E^2R}} \\
l = 2\sqrt[3]{\dfrac{3WR}{4E}} \\
\dfrac{1}{E} = \dfrac{1-v_1^2}{E_1} + \dfrac{1-v_2^2}{E_2}
\end{cases}
\tag{2}
$$

where $W$ is the external load, $\delta$ is the normal deformation depth, $l$ is the diameter of the real contact area, and $R$ is the curvature radius of the peak. $E$ is the equivalent modulus, and $E_1$ and $E_2$ are Young's moduli of the friction pair. $v_1$ and $v_2$ are Poisson's ratios of the friction pair, respectively.

According to Equation (2), the geometric relationship between the real contact area of the peak and the normal deformation depth can be obtained, that is:

$$
a_r = \frac{\pi l^2}{4} = \pi R \delta
\tag{3}
$$

where $a_r$ is the real contact area. However, the curvature radius of the peak changes with the contact state during the deformation process, and the size of the peak is related to the real contact area. Therefore, the relationship between the real contact area and the normal deformation depth can be established based on the fractal theory. It can be seen from the W–M function that the roughness is the peak height of the rough surface with the cosine wave characteristic, which can be defined as [31], that is:

$$
\begin{cases}
z_{l(x,y)} = G^{D_s-1} l^{2-D_s} \cos\left[\dfrac{\pi l(x,y)}{l}\right] \\
-\dfrac{l}{2} < l(x,y) < \dfrac{l}{2}
\end{cases}
\tag{4}
$$

where $D_s$ is the 2-D fractal dimension, $D_s = D-1$. Substituting Equation (3) into Equation (4), the relationship between the normal deformation depth of the peak and the real contact area can be obtained, that is:

$$
\delta = G^{D_s-1} l^{2-D_s} = G^{D_s-1}\left(\frac{4a_r}{\pi}\right)^{\frac{2-D_s}{2}}
\tag{5}
$$

Then, the real contact area of a single peak during the contact process can be expressed as:

$$
a_r = \frac{\pi}{4}\left(\frac{\delta}{G^{D_s-1}}\right)^{\frac{2}{2-D_s}}
\tag{6}
$$

By summing the real contact areas of all peaks on the rough surface, the total real contact area of the rough surface can be obtained as follows:

$$A_r = \sum a_r \ (a_{\min} \leq a_r \leq a_{\max}) \tag{7}$$

where $A_r$ is the total real contact area of the rough surface, $a_{\min}$ is the minimum real contact area of a single peak, and $a_{\max}$ is the maximum real contact area of a single peak.

### 2.2. Wear Calculation Based on Fractal Theory

The rough surfaces in contact with each other would inevitably wear during the friction process. The factors affecting the wear amount include the relative sliding distance of the friction pair, the external load, and the properties of the friction material. Archard summarized the factors affecting the wear phenomenon and proposed a wear calculation model [32]:

$$V = K\frac{W}{\sigma_s}S \tag{8}$$

where $V$ is the wear volume of the worn material, $K$ is the wear coefficient of the friction pair, $S$ is the relative sliding distance of the friction pair, and $\sigma_s$ is the yield limit of the worn material. Considering the elastic deformation and plastic deformation of the rough surface in the friction process, the fractal wear calculation model expressed by fractal parameters, working condition parameters, and material performance parameters is established [33], that is:

$$V = \left(1 + \lambda\mu^2\right)^{\frac{1}{2}} \cdot a_r \cdot \left\{ k_{we} + (k_{wp} - k_{we}) \left[ \frac{D_s G^2 \sigma_y}{(2 - D_s)W} \left( \frac{\pi E^2}{225\sigma_s^2} \right)^{\frac{1}{D_s - 1}} \right]^{\frac{2 - D_s}{2}} \psi^{\frac{(D_s - 2)^2}{4}} \right\} \cdot v \cdot T \tag{9}$$

where $\lambda$ is an experimental constant [34], $\mu$ is the friction coefficient, $k_{we}$ is the elastic wear coefficient, $k_{wp}$ is the plastic wear coefficient, $\sigma_y$ is the strength limit of material deformation, $\psi$ is the domain expansion factor, $v$ is the relative sliding speed, and $T$ is the friction time.

Assuming that the nominal contact area of the rough surface is $a_n$, the average wear depth $h$ of the rough surface can be expressed as:

$$h = \left(1 + \lambda\mu^2\right)^{\frac{1}{2}} \cdot \frac{a_r}{a_n} \cdot \left\{ k_{we} + (k_{wp} - k_{we}) \left[ \frac{D_s G^2 \sigma_y}{(2 - D_s)W} \left( \frac{\pi E^2}{225\sigma_s^2} \right)^{\frac{1}{D_s - 1}} \right]^{\frac{2 - D_s}{2}} \psi^{\frac{(D_s - 2)^2}{4}} \right\} \cdot v \cdot T \tag{10}$$

Substituting Equation (6) into Equation (10), the wear calculation equation can be obtained:

$$h = \left\{ k_{we} + (k_{wp} - k_{we}) \left[ \frac{D_s G^2 \sigma_y}{(2 - D_s)W} \left( \frac{\pi E^2}{225\sigma_s^2} \right)^{\frac{1}{D_s - 1}} \right]^{\frac{2 - D_s}{2}} \psi^{\frac{(D_s - 2)^2}{4}} \right\} \cdot \left(1 + \lambda\mu^2\right)^{\frac{1}{2}} \cdot \frac{1}{a_n} \cdot \frac{\pi}{4} \left( \frac{\delta}{G^{D_s - 1}} \right)^{\frac{2}{2 - D_s}} \cdot v \cdot T \tag{11}$$

Considering the contact state of a single peak, the average wear depth of the surface with rough topography can be estimated based on Equation (11).

### 2.3. Digital Image Conversion of Rough Surfaces

The real engineering surface is a continuous rough surface. To numerically analyze the rough surface, combined with the concept of calculus, the continuous rough surface is discretized based on ensuring the topographic characteristics of the rough surface. The appropriate element size of the discrete element is selected, and the topographic information of the rough surface is expressed in the form of 3-D data points. As shown in Figure 3, a rough surface with fractal characteristics is established based on the functional equation

(Equation (1)), $x_i$ and $y_j$ are used to represent the coordinate points of the discrete unit, $z(x_i,y_j)$ is the height value of the discrete unit, and then the discrete unit data point is recorded as $H[x_i,y_j,z(x_i,y_j)]$, which realizes the transformation of the real rough surface to the virtual data surface, that is, the digital processing of the image.

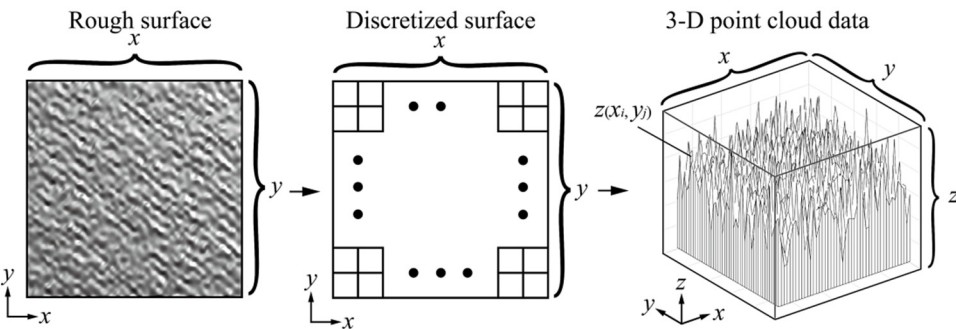

**Figure 3.** Digital image conversion of rough surfaces.

The fractal dimension $D$ of the 3-D space has a certain relationship with the fractal dimension $D_s$ of the 2-D plane, that is, $D = D_s + 1$. Therefore, the information on the 3-D morphology can be stored in the 2-D plane, and the 3-D fractal dimension can be calculated by measuring the 2-D fractal dimension of the 2-D plane. Based on image recognition technology, the fractal dimension of the rough surface is calculated, and the 3-D coordinate data are converted into image color data. The 3-D data point set $H[x_i,y_j,z(x_i,y_j)]$ of the rough surface is represented by the 2-D data point set group $P(x_i,y_j)$ and $g(x_i,y_j)$, where $P(x_i,y_j)$ is the coordinate of the discrete unit, $g(x_i,y_j)$ is the gray value of the discrete unit, and $g(x_i,y_j)\in(0.255)$. Therefore, the 3-D space information can be represented by a 2-D plane grayscale image, as shown in Figure 4. Furthermore, the change in the rough surface morphology during the wear process can be expressed by the grayscale change of the grayscale image. In addition, the grayscale image not only contains the plane position information, but can also store the color grayscale data. Therefore, the use of the gray scale image is convenient to measure the fractal dimension of the two-dimensional surface with MATLAB, and then to express the fractal dimension of rough surface with three-dimensional information.

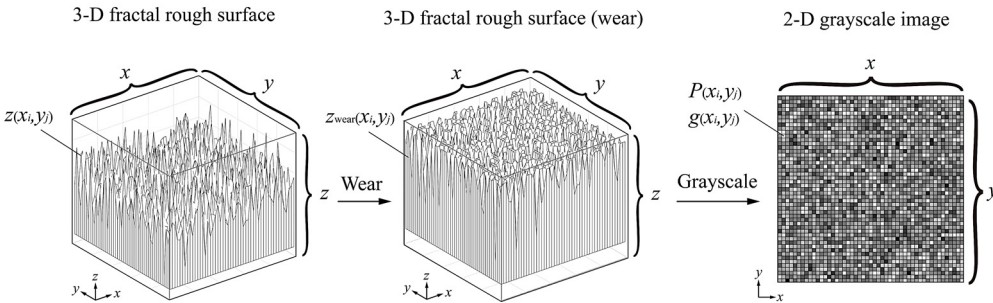

**Figure 4.** Conversion of the rough surface from 3-D to 2-D.

### 2.4. Time-Varying Wear Calculation Method

The distribution of the wear degree in the friction area of the rough surface is different. Considering the friction contact state of the initial surface, combined with the analysis of the friction contact mechanism of the rough surface, the rigid plane is discretized, and the overall rigid plane is transformed into an $x \times y$ rigid element plane. Thus, the wear problem between the rigid plane and the rough surface is transformed into the friction contact problem between the rigid element plane and the peak, as shown in Figure 5.

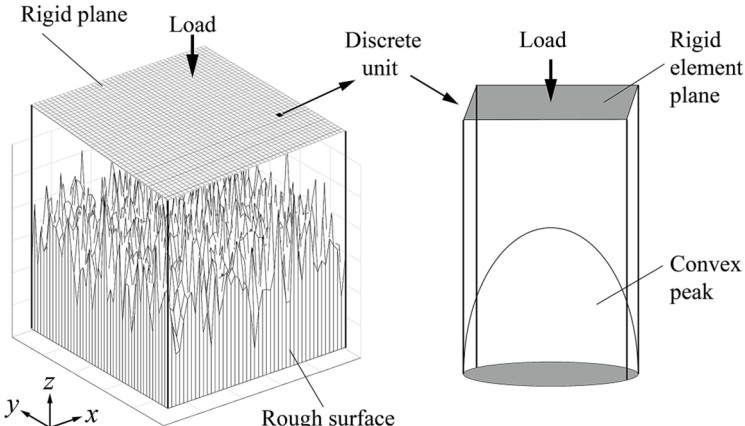

**Figure 5.** Rough surface discretization characterization.

In the friction process, the wear criterion for the rough surface in unit time is whether the peak is deformed by friction contact, as well as whether the contact stress reaches the yield limit of the material. As shown in Figure 6, $z_1$ is the peak height that will wear, $z_2$ is the peak height that will not wear, and $z_3$ is the peak height equal to the wear boundary. It is assumed that the surface wear depth per unit time is $\Delta h_{\text{wear}}$, and the wear depth of discrete units is $\Delta h_{\text{wear}}(x_i,y_j)$. The peak value of the unworn rough surface is $z(x_i,y_j)$, the maximum peak value of the rough surface is $z_{\text{max}}$ and the minimum peak value of the rough surface is $z_{\text{min}}$. The maximum peak value of the worn rough surface is $z_{\text{wearmax}}$, that is:

$$\begin{cases} z_{\text{max}} = \Delta h_{\text{wear}} + \max\left[z(x_i, y_j) - \Delta h_{\text{wear}}(x_i, y_j)\right] \\ z_{\text{wearmax}} = \max\left[z(x_i, y_j) - \Delta h_{\text{wear}}(x_i, y_j)\right] \end{cases} \tag{12}$$

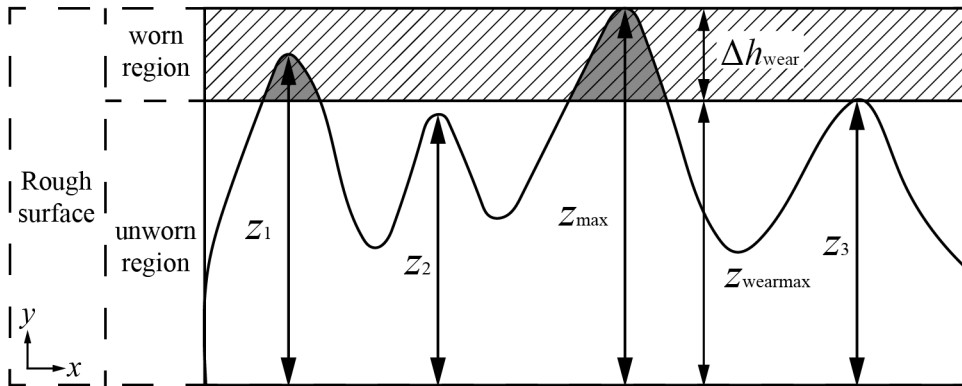

**Figure 6.** Surface topography changes of peaks during friction.

When the rigid plane and the rough surface are in friction contact, the rigid plane and the highest peak of the rough surface contact first. After a unit time $\Delta T$, the wear phenomenon of the highest peak occurs, and the contact states of peaks are changed and different. Therefore, the topographic changes of the rough surface are divided into two types: material loss caused by wear of peaks; deformation of peaks caused by friction.

(1)    Time-varying wear calculation model of the worn surface

When the peak is in a friction contact state, the peak value of the rough surface is greater than the worn peak value, it is specified that material wear occurs, and the contact peak value is the maximum peak value of the worn rough surface, that is:

$$\begin{cases} z_{\text{wear1}} = z_{\text{wearmax}}(z_1 > z_{\text{wearmax}}) \\ z_{\text{wear3}} = z_{\text{wearmax}}(z_3 = z_{\text{wearmax}}) \end{cases} \tag{13}$$

where $z_{\text{wear1}}$ is the peak height of $z_1$ after wear calculation and $z_{\text{wear3}}$ is the peak height of $z_3$ after wear calculation. Meanwhile, the deformation value of the peak can be expressed by $\delta_{\text{wear}}(x_i, y_j)$, that is:

$$\delta_{\text{wear}}(x_i, y_j) = z(x_i, y_j) - z_{\text{wearmax}} \tag{14}$$

Then, the calculation equation of the wear depth of the contact peak is:

$$\Delta h_{\text{wear}}(x_i, y_j) = \left\{ k_{we} + (k_{wp} - k_{we}) \left[ \frac{D_s G^2 \sigma_y}{(2-D_s)W} \left( \frac{\pi E^2}{225\sigma_s^2} \right)^{\frac{1}{D_s-1}} \right]^{\frac{2-D_s}{2}} \psi^{\frac{(D_s-2)^2}{4}} \right\} \cdot \\ (1 + \lambda\mu^2)^{\frac{1}{2}} \cdot \frac{1}{a_n} \cdot \frac{\pi}{4} \left[ \frac{z(x,y)-z_{\text{wearmax}}}{G^{D_s-1}} \right]^{\frac{2}{2-D_s}} \cdot v \cdot T \tag{15}$$

(2)   Compensation wear calculation model of the unworn surface

When the peak is in a friction non-contact state, that is, the peak value of the rough surface is smaller than the worn peak value, then:

$$z_{\text{wear2}} = z_2 \; (z_2 < z_{\text{wearmax}}) \tag{16}$$

where $z_{\text{wear2}}$ is the peak height of $z_2$ after wear calculation.

However, non-contact peaks are affected by temperature and external conditions on the friction surface. Therefore, considering the friction characteristics of the rough surface, 0.1% and 0.2% of the maximum peak-to-pit distance $R_{\max}$ of the initially rough surface are taken as the initial deformation $\delta_0$ of the peak, which is expressed as follows:

$$\begin{cases} \delta_0 = R_{\max} \cdot \text{rand}[0.001, 0.002] \\ R_{\max} = z_{\max} - z_{\min} \end{cases} \tag{17}$$

Then, the calculation equation of the wear depth of the non-contact peak is:

$$\Delta h_{\text{wear}}(x_i, y_j) = \left\{ k_{we} + (k_{wp} - k_{we}) \left[ \frac{D_s G^2 \sigma_y}{(2-D_s)W} \left( \frac{\pi E^2}{225\sigma_s^2} \right)^{\frac{1}{D_s-1}} \right]^{\frac{2-D_s}{2}} \psi^{\frac{(D_s-2)^2}{4}} \right\} \cdot \\ (1 + \lambda\mu^2)^{\frac{1}{2}} \cdot \frac{1}{a_n} \cdot \frac{\pi}{4} \left[ \frac{(z_{\max}-z_{\min}) \cdot \text{rand}[0.001,0.002]}{G^{D_s-1}} \right]^{\frac{2}{2-D_s}} \cdot v \cdot T \tag{18}$$

The peak value $z_{\text{wear}}$ of the worn rough surface can be expressed as:

$$z_{\text{wear}}(x_i, y_j) = \begin{cases} z_{\max} - \Delta h_{\text{wear}} & [z(x_i, y_j) - \Delta h_{\text{wear}}(x_i, y_j) \geq z_{\max} - \Delta h_{\text{wear}}] \\ z(x_i, y_j) - \Delta h_{\text{wear}}(x_i, y_j) & [z(x_i, y_j) - \Delta h_{\text{wear}}(x_i, y_j) < z_{\max} - \Delta h_{\text{wear}}] \end{cases} \tag{19}$$

The discrete unit data points of the worn surface are recorded as $H_{\text{wear}}[x_i, y_j, z_{\text{wear}}(x_i, y_j)]$, and a new 3-D rough surface can be constructed. To comprehensively explore the surface wear mechanism of the friction pair, it is not only necessary to calculate the wear depth of the surface, but also to analyze the change laws of the parameters representing the surface topography of the friction pair. Usually, the surface topography characterization parameters are expressed by the surface height deviation $S_a$, the root-mean-square deviation $S_q$, and the surface kurtosis $S_{ku}$. The calculation equations are:

$$u = \frac{1}{MN} \sum_{i=1}^{M} \sum_{j=1}^{N} z(x_i, y_j) \tag{20}$$

$$S_a = \frac{1}{MN} \sum_{i=1}^{M} \sum_{j=1}^{N} |z(x_i, y_j) - u| \tag{21}$$

$$S_q = \sqrt{\frac{1}{MN}\sum_{i=1}^{M}\sum_{j=1}^{N}[z(x_i,y_j)-u]^2} \tag{22}$$

$$S_{ku} = \frac{1}{MNS_q^4}\sum_{i=1}^{M}\sum_{j=1}^{N}[z(x_i,y_j)-u]^4 \tag{23}$$

where $M$ and $N$ are the number of data points.

Based on the differential box-counting method [35], the Fraclab Toolbox module in MATLAB is used to identify and calculate the fractal dimension of the worn surface image, and then to calculate the wear depth of the rough surface according to the wear equations. The rough surface topography is updated, and the worn surface topography is identified and calculated again to realize the iterative calculation of the wear cycle, and to obtain the changing trend and wear laws of the rough surface during the wear process. The specific process is as follows:

(1) The rough surface is established by input basic parameters, and the rough surface is characterized by the data points set.
(2) The appropriate time interval is selected to calculate the wear depth of the worn rough surface.
(3) The point data of the worn surface are calculated by the wear judgment criterion, and the worn rough surface is established.
(4) Through the image digital processing method, the 3-D topography is converted into a 2-D grayscale image and the fractal dimension of the worn rough surface is calculated.
(5) The cycle calculation time is set and the surface wear depth and surface topography characterization parameters are output. The flowchart is shown in Figure 7.

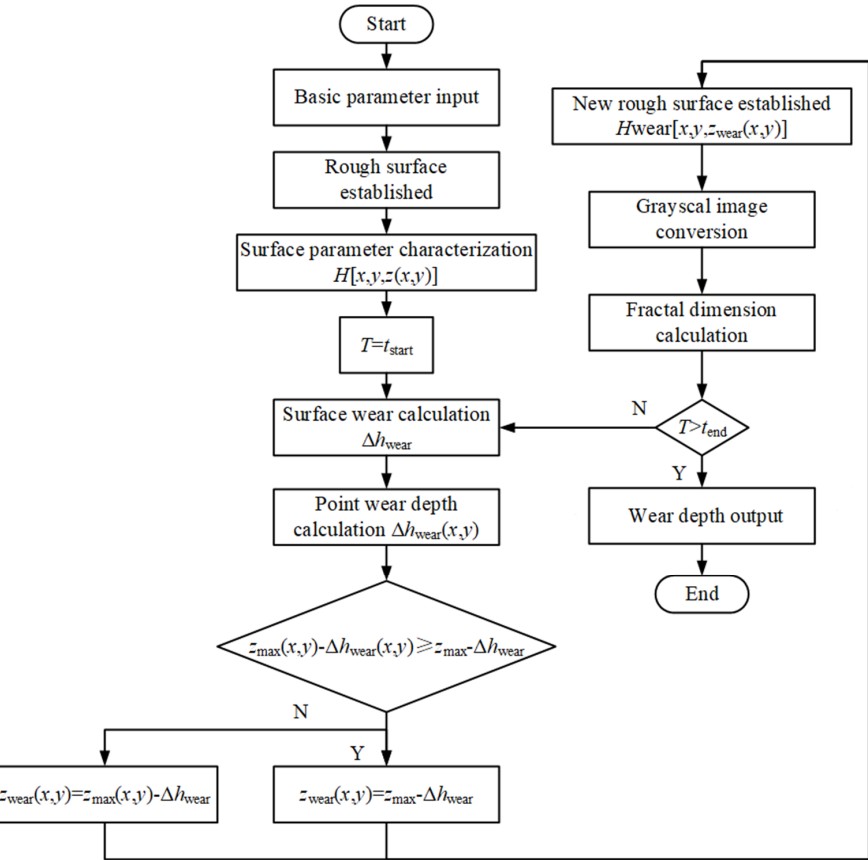

**Figure 7.** Flow chart of iterative calculation of wear cycle.

## 3. Results of Simulation and Experiment

### 3.1. Numerical Simulation Analysis

The pin-on-disc friction pair is taken as the friction object, and the wear numerical simulation on the rough surface of the friction pair is carried out by using MATLAB software (version R2016b). The sample pin is 4.8 mm in diameter and 12 mm in length. The inner diameter of the sample disc is 38 mm, the outer diameter is 54 mm, the friction radius is 23 mm, and the thickness is 10 mm, as shown in Figure 8. The friction parameters of the sample disc and the sample pin are shown in Table 1.

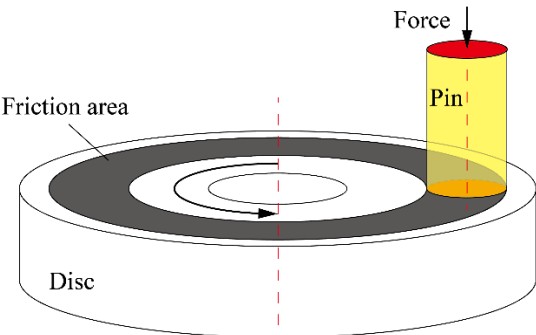

**Figure 8.** Schematic diagram of the pin-on-disc friction pair.

**Table 1.** Friction parameters of the friction pair.

| Items | Numerical Value | Items | Numerical Value |
|---|---|---|---|
| $E_1/(\text{MPa})$ | $1.8 \times 10^5$ | $E_2/(\text{MPa})$ | $2.04 \times 10^5$ |
| $\nu_1$ | 0.3 | $\nu_2$ | 0.31 |
| $k_{we}$ | $1 \times 10^{-6}$ | $k_{wp}$ | $1 \times 10^{-3}$ |
| $\lambda$ | 9 | $\mu$ | 0.3 |
| $\sigma_y/(\text{MPa})$ | 345 | $D_0$ | 2.290 |

The friction parameters are variable at rotational speeds of 500 r/min, 600 r/min, 700 r/min, 800 r/min, 900 r/min, and 1000 r/min, with external loads of 30 N, 50 N, 70 N, 90 N, 110 N, and 130 N, and the control variable method is used to explore the influence of speed and load on the tribological properties of the friction pair, as shown in Table 2. In addition, in the process of the pin-on-disc wear experiment, due to the influence of friction parameters, the temperature change of the friction pair surface is small, so this paper ignores the influence of temperature on the friction pair surface wear under this friction condition.

**Table 2.** Friction parameters of friction conditions.

| No. | $W$/(N) | $v$/(r/min) | No. | $W$/(N) | $v$/(r/min) | No. | $W$/(N) | $v$/(r/min) |
|---|---|---|---|---|---|---|---|---|
| 1 | 30 | 500 | 10 | 70 | 900 | 19 | 110 | 800 |
| 2 | 50 | 600 | 11 | 70 | 1000 | 20 | 110 | 900 |
| 3 | 50 | 700 | 12 | 90 | 600 | 21 | 110 | 1000 |
| 4 | 50 | 800 | 13 | 90 | 700 | 22 | 130 | 600 |
| 5 | 50 | 900 | 14 | 90 | 800 | 23 | 130 | 700 |
| 6 | 50 | 1000 | 15 | 90 | 900 | 24 | 130 | 800 |
| 7 | 70 | 600 | 16 | 90 | 1000 | 25 | 130 | 900 |
| 8 | 70 | 700 | 17 | 110 | 600 | 26 | 130 | 1000 |
| 9 | 70 | 800 | 18 | 110 | 700 | - | - | - |

The simulation operation takes the rotational speed of 500 r/min, the external load of 30 N, and the wear time of 60 min as examples. Combined with the mathematical model of

the time-varying wear calculation, the numerical wear calculation is carried out by virtual simulation MATLAB software (version R2016b), and the topographical changes of the rough surface before and after wear can be obtained, as shown in Figure 9.

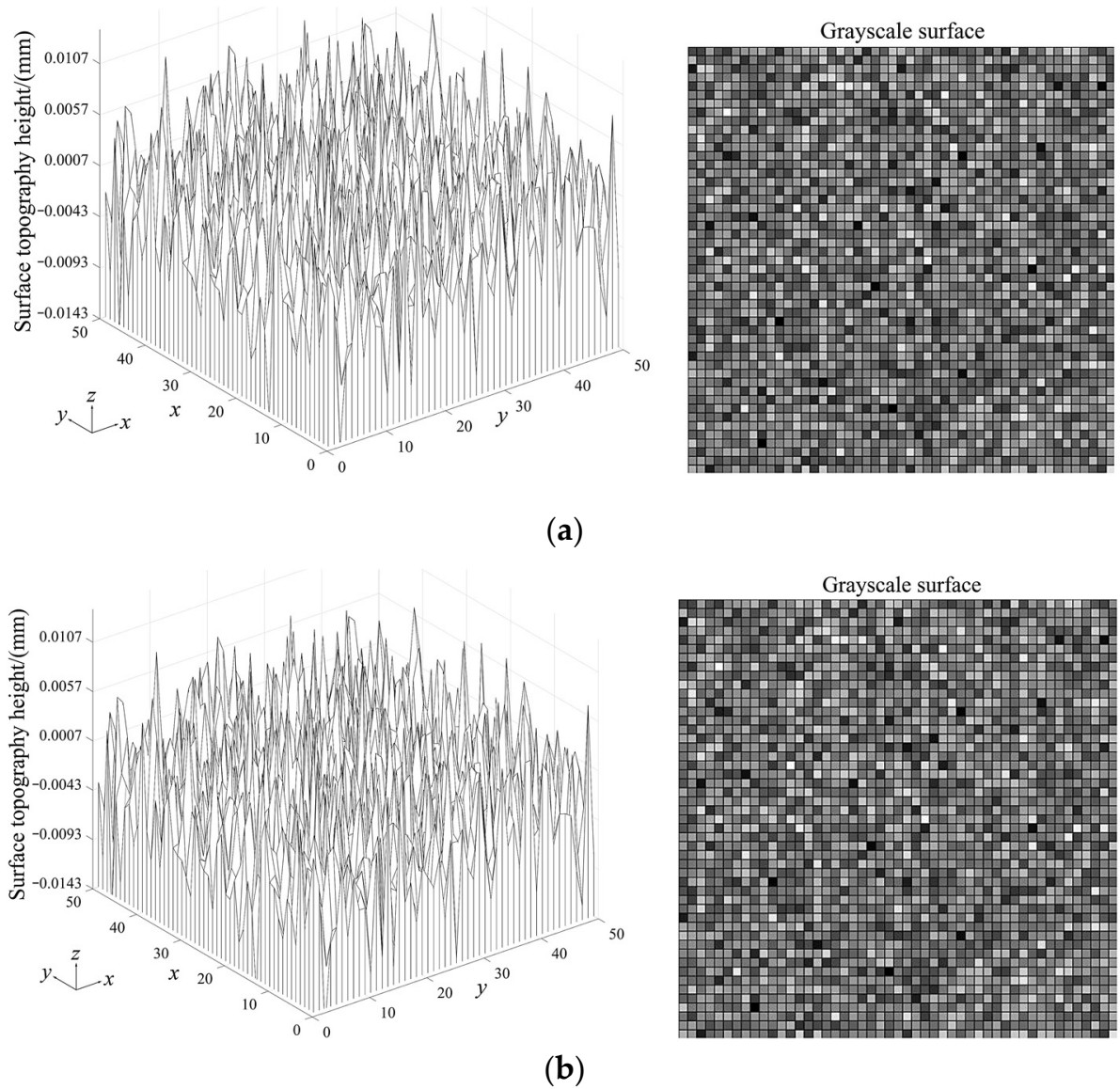

**Figure 9.** Wear cycle iterative calculation flowchart. (**a**) Surface topography when friction time is 0 min; (**b**) surface topography when friction time is 60 min.

During the wear process, the rough surface would be squeezed and sheared, and the surface topography changes after the wear compared with the primary surface topography. The wear phenomenon of rough surfaces occurs first on high peaks. Under the action of the external force, when the bearing capacity of the peak exceeds its strength limit, the peak would be destroyed and damaged, resulting in the wear of the rough surface. However, the pit areas are not in contact and do not wear. Moreover, in the actual surface wear process, the pit areas may be filled and compacted by wear debris, or surface defects may be generated due to plastic deformation or scribing. Therefore, in the process of wear calculation, surface defect compensation is performed on the unworn rough peaks, to simulate the influence of other factors on the unworn surface. In addition, from the comparison of the 2-D grayscale images of the unworn surface and the worn surface, it can be seen that the overall structure and surface texture of the grayscale results remain

relatively stable, and the fractal features are the same, indicating that the rough surface is correlated before and after wear.

The bearing surface curves can be used to represent the distribution of peaks on the surface profile in the height direction and can reflect the area size of the bearing surface area of the rough surface during the wear process. It is considered to be an effective method for describing surface profiles and plays an important role in studying the contact state and surface wear resistance of friction surfaces.

Through the statistics of the peak height of the surface topography before and after wear, the bearing surface curves can be drawn, as shown in Figure 10. To facilitate the comparative analysis of change laws of the surface topography, the surface topography heights are divided into three parts: surface topography heights with a height value greater than 0.005 mm are defined as a peak region, surface topography heights with a height value of less than −0.01 mm are defined as a pit region, and surface topography heights of 0.01~0.005 mm are defined as the middle region. The overall peak height of worn rough surfaces tends to decrease. The ratio of the peak region decreases from 16.26% to 4.39%, and the peak value decreases significantly. The ratio of the pit region increases from 3.32% to 17.13%, and the peak value also increases greatly. The ratio of the middle region decreases from 80.42% to 78.48%, with a small decrease in the peak value. It shows that in the wear process, plastic deformation and material damage occur in the peak region of the rough surface, which increases the wear degree of the surface. The peak value of the worn surface in the peak region decreases, and the ratio of the middle area is increased. At the same time, the plastic deformation of peaks leads to an increase in the real contact area. Furthermore, the number of peaks is evenly distributed, and the pressure on the contact peak is reduced. The shearing and squeezing effects between the friction pairs are weakened, and the bearing capacity of the surface is improved.

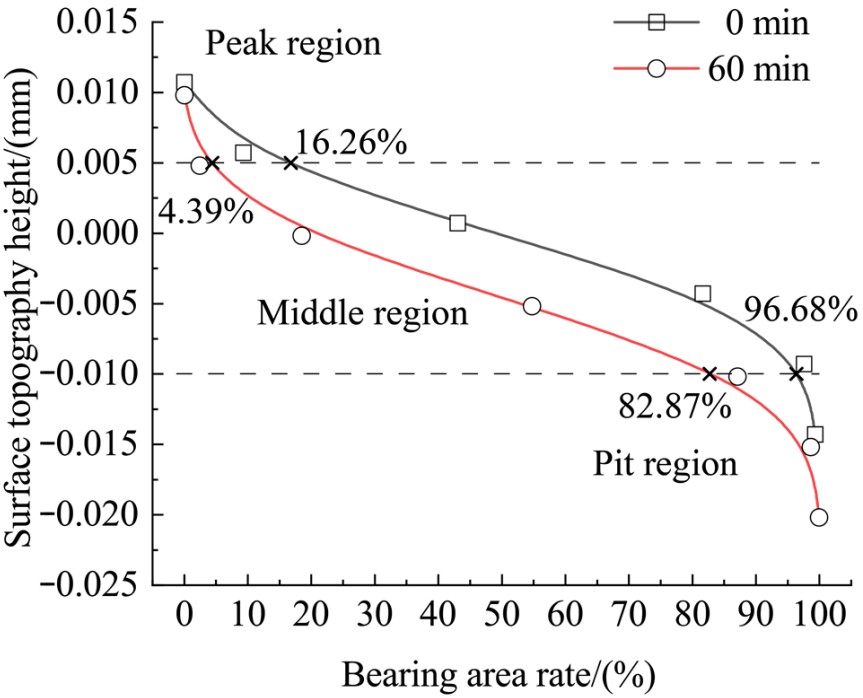

**Figure 10.** Bearing area curves of surface topography.

### 3.2. Pin-on-Disc Wear Experiment Pretreatment

The experimental equipment used in the pin-on-disc wear experiment is the MMW-1A microcomputer-controlled universal friction and wear tester (Jinan Yihua Tribology Testing Technology Co., Ltd, Jinan, China), as shown in Figure 11. The experimental machine is mainly composed of an electric control panel, spindle drive system, spring loading

system, embedded computer measurement and control system, friction pair, and special fixtures. The whole experimental process can be controlled through the computer, and the system software can monitor the experimental process in real-time and record the relevant experiment parameters and curves. The working principle is that the servo motor drives the main shaft through the synchronous pulley, and the sample disc and fixture are connected with the main shaft through the tie rod and rotate synchronously with the main shaft. It has the advantages of an accurate transmission ratio, high control precision, low noise, and no pollution.

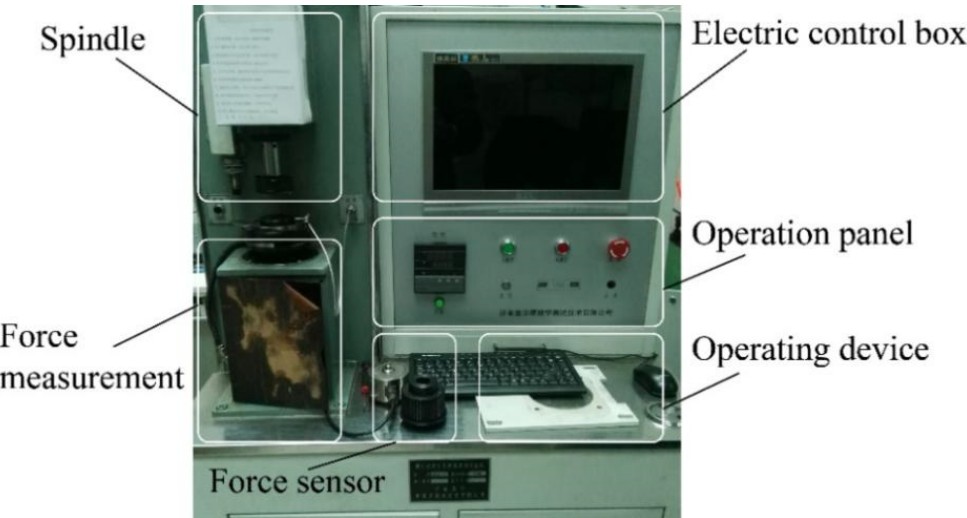

**Figure 11.** Pin-on-disc wear experimental machine.

The pin-on-disc wear experiment adopts the sample pin and the sample disc with the dimensions shown in Figure 8, and the material properties are shown in Table 1. The experimental conditions are the same as the simulation conditions. During testing, the room temperature is 20 °C and the relative humidity is 25%~50%. Before and after the experiment, the sample pin and the sample disc are washed with absolute ethanol and dried. The surface topography before and after wear is measured by an optical 3-D surface topography measuring instrument (BRUKER ALICONA CHINA, Beijing, China), and the changes in rough surface topography and surface roughness before and after wear are obtained, which are compared with the numerical simulation results.

*3.3. Comparative Analysis of Wear Results*

Through the pin-on-disc wear experiment, the surface topography changes before and after wear can be obtained. The contour curves of the surface topography are extracted and the effect of wear on the change in surface topography is analyzed, as shown in Figure 12. From the contour changes of the surface topology, it can be seen that there is a significant peak height difference in the unworn surface topography, and there are both peak areas and pit areas. After 60 min of wear time, the surface topography height is reduced, of which the peak height in the peak region is significantly reduced. This is because the rough surface is the first to contact the peak region during the contact process and is in a frictional state, causing the material to be damaged and cause wear. However, with the reduction in the surface topography height in the pit region, the contact state never changes to the actual contact wear state, and the pit depth becomes smaller. On the whole, while the height of the surface topography is decreasing, the height difference between the highest peak and the lowest peak becomes small, and the surface roughness is reduced, which increases the surface contact area and strengthens the surface-bearing capacity.

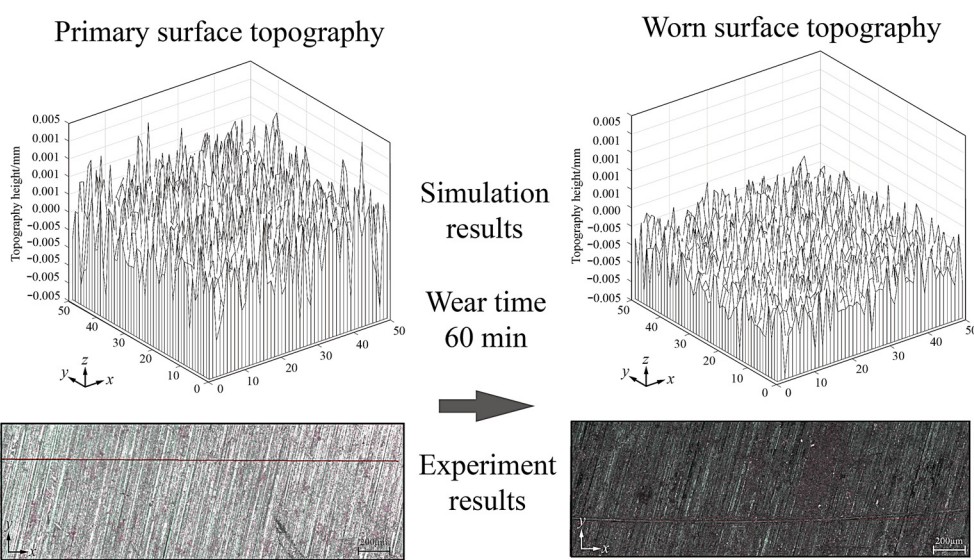

**Figure 12.** Changes in the surface topography during the wear process.

The experimental data and simulation data are extracted, and the height of the rough surface topography is sampled and analyzed five times. Furthermore, the average height value of the surface topography is calculated, and the difference between the average height before and after wear is used as the average wear depth, as shown in Figure 13. As for the experimental results, the average height of the unworn surface topography is $1.51 \times 10^{-3}$ mm, the average height of the worn surface topography is $-2.03 \times 10^{-2}$ mm, and the average wear depth is $2.18 \times 10^{-2}$ mm. Meanwhile, as for the simulation results, the average height of the unworn surface topography is $8.69 \times 10^{-4}$ mm, the average height of the worn surface topography is $-2.14 \times 10^{-2}$ mm, and the average wear depth is $2.22 \times 10^{-2}$ mm. By comparing the wear simulation and wear experiment results, it can be seen that the maximum profile height of the surface is significantly reduced during the wear process, and the error of the average height value of the surface topography is 1.83%, which proves the effectiveness of the wear simulation.

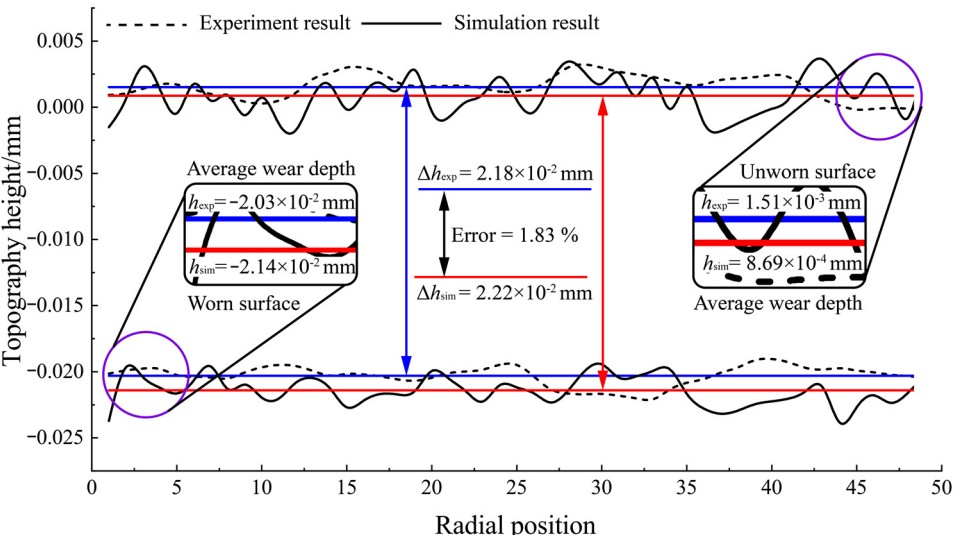

**Figure 13.** Changes in the wear depth during the wear process.

## 4. Discussion and Analysis of Surface Topography Parameters of Friction Pairs

The wear of the material is the interaction of the friction pair, which causes the surface material to deform and destroy. The performance parameters of the friction pair material

affect the friction and wear performance of the friction pair surface. The state and change of surface topography characteristics can directly reflect the performance of friction and wear of materials. Therefore, reasonable characterization of surface topographic features is an effective method to study the friction and wear properties of materials.

### 4.1. Effect of Friction Parameters on Surface Wear Depth

The surface wear of the friction pair is related to the friction parameters, and the rotational speed and external load would affect the result of the surface wear depth, as shown in Figure 14. When the initial friction parameters are the initial rotational speed of 1000 r/min and the external load of 130 N, the maximum wear depth on the surface of the friction pair is 99.3 μm. With the decrease in the initial rotational speed and the external load, the wear depth gradually decreases, and the decreasing trend shows a gradient distribution. When the initial friction parameters are the initial rotational speed of 600 r/min and the external load of 50 N, the minimum wear depth on the surface of the friction pair is 28.7 μm.

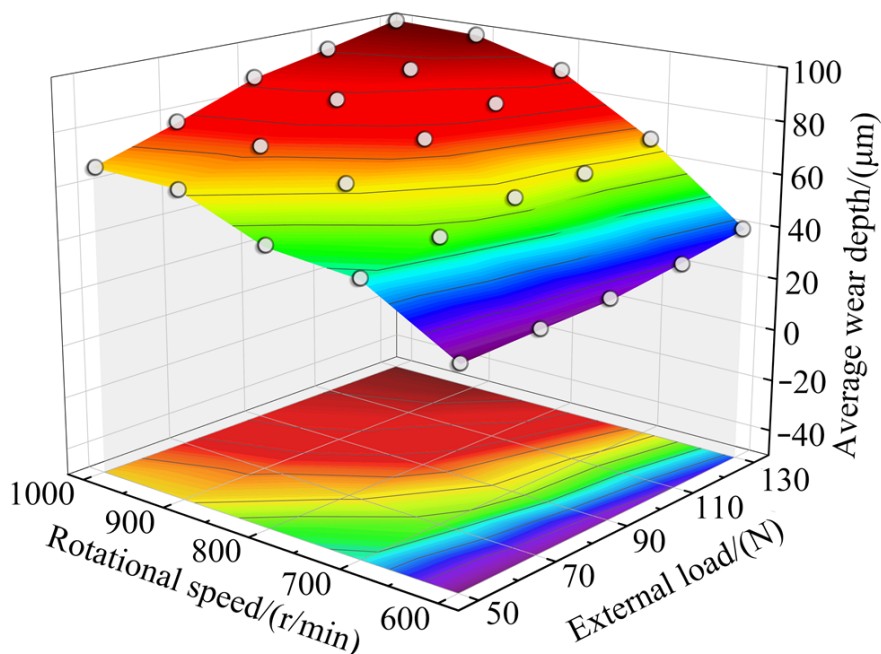

**Figure 14.** Effect of friction parameters on surface wear depth. (Different colors represent the difference in average wear height, dark red represents the maximum value, and purple represents the minimum value.)

When the external load is increased from 50 N to 130 N, the external load change has a small impact on the surface wear of low rotational speed (600 r/min~800 r/min), its wear depth increases by 39.02%, the impact on the surface wear of high rotational speed (800 r/min~1000 r/min) is large, and the wear depth increases by 48.88%. The large load would increase the normal pressure on the surface, causing plastic deformation and ductile fracture of the surface peaks. Therefore, when the nominal contact area is the same, the number of wear peaks increases with the external load, so the total wear depth increases at the end of friction time. Meanwhile, Yang and Jin found that the surface material loss increased with the increase in friction load [36]. When the initial rotational speed increased from 600 r/min to 1000 r/min, the high rotational speed not only increased the friction area per unit time but also increased the probability of friction extrusion of the peak. Therefore, at the end of friction time, the increase in the total friction area would make the total wear depth grow, which, in turn, causes the surface wear to rise with the increase in rotational speed.

In addition, the accuracy of the wear calculation model is analyzed by comparing the calculation results of the literature [37]. In order to study the wear law obtained from the wear model, the wear data in the literature [37] and the wear data in this paper are normalized, and the effect of friction load on the surface wear law is analyzed, as shown in Figure 15. It can be seen from the figure that the increase in friction load can increase the surface wear. Compared with the wear depth calculation curve in the literature, the trend of the wear depth curve calculated in this paper is close to the wear law obtained from the test. This is because there are many factors considered in this paper.

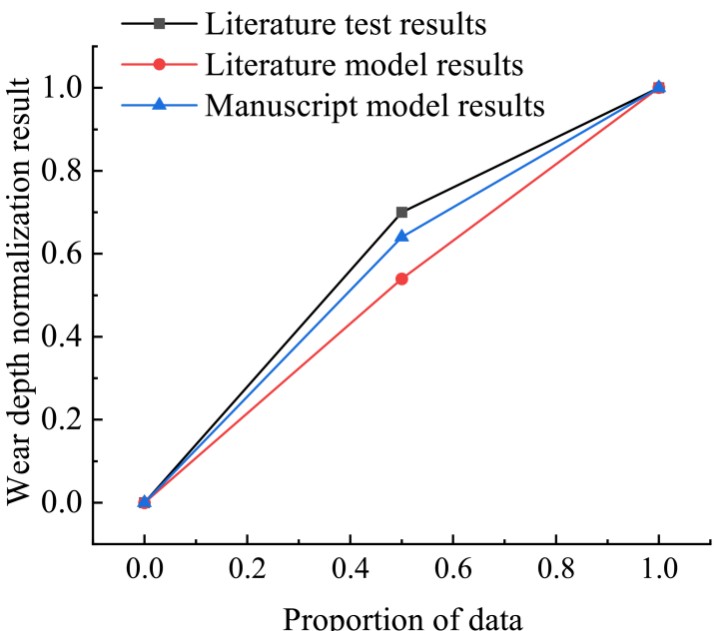

**Figure 15.** Comparison and analysis of normalization results.

### 4.2. Effect of Friction Parameters on the Surface Topography and Characteristics

The surface height deviation $S_a$ reflects the arithmetic average distribution of the surface roughness height information, which can effectively reflect the height characteristics of the overall sample selection area. As shown in Figure 16, it is a graph showing the relationship between the surface height deviation $S_a$ as a function of the external load and the initial rotational speed. When the external load increases from 50 N to 130 N, the value of $S_a$ rises with an average increase of 17.82%. This is because the large external load increases the interaction of the contact surfaces, and the large friction force increases the surface wear and improves the surface roughness, that is, the surface height deviation $S_a$ is raised with the increase in the external load. In addition, when the surface topography parameters are large, the Hertzian stress can increase and cause the surface to be deformed [18]. At low rotational speed, the value of $S_a$ increases by 2.93% with the increase in the rotational speed, and the wear depth per unit time is small. At this time, the main form of wear is surface scratching. Therefore, the peak value of the worn peak is reduced, and the pit region peak value remains unchanged, so the surface height deviation $S_a$ is small and the magnitude of the increase is not obvious. At high rotational speed, the value of $S_a$ increases by 3.88% with the increase in the rotational speed, the wear depth per unit time is large, and the surface wear effect is strengthened. The pit region is affected by the abrasive wear to produce the phenomenon of surface scratching, which reduces the surface height and increases the surface roughness, that is, the surface height deviation $S_a$ increases with the increase in the rotational speed.

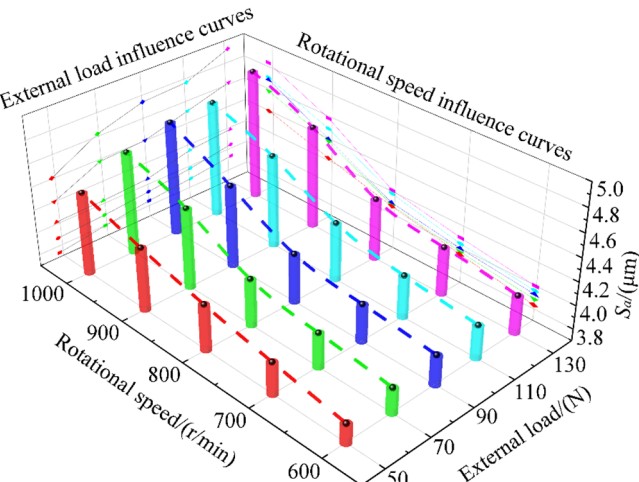

**Figure 16.** Effect of friction parameters on surface height deviation. (red is the value of $S_a$ at different rotational speeds at 50N, green is the value of $S_a$ at different rotational speeds at 70N, blue is the value of $S_a$ at different rotational speeds at 90N, light blue is the value of $S_a$ at different rotational speeds at110N, and pink is the value of $S_a$ at different rotational speeds at 130N).

The root-mean-square deviation $S_q$ of the surface height distribution can reflect the root-mean-square value of the deviation distance between the microscopic surface and the rated datum within the primary surface area. As shown in Figure 17, it is a graph showing the relationship between the root-mean-square deviation $S_q$ of the surface height distribution and the variation in the external load and the rotational speed. As the external load is reduced from 130 N to 50 N, $S_q$ decreases by 3.06% at a low rotational speed and 7.05% at a high rotational speed. When the rotational speed reduces from 1000 r/min to 600 r/min, $S_q$ decreases by 17.47% with the decrease in the rotational speed, and the value decreases significantly. The value of $S_q$ is small, the microscopic surface is smooth, and the stability of the surface coordination is well.

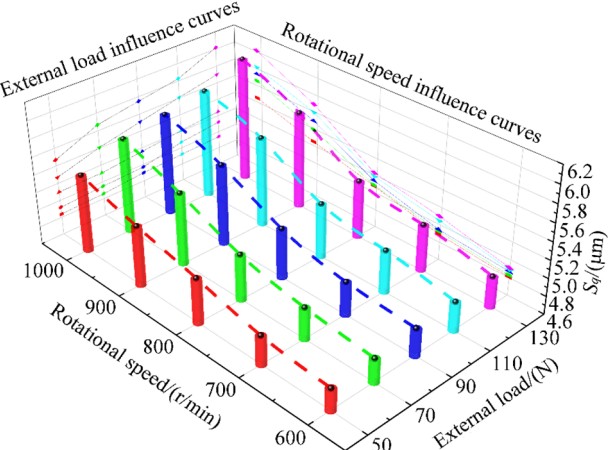

**Figure 17.** Effect of friction parameters on the root-mean-square deviation. (red is the value of $S_q$ at different rotational speeds at 50N, green is the value of $S_q$ at different rotational speeds at 70N, blue is the value of $S_q$ at different rotational speeds at 90N, light blue is the value of $S_q$ at different rotational speeds at110N, and pink is the value of $S_q$ at different rotational speeds at 130N).

The surface kurtosis $S_{ku}$ describes the shape of the height distribution of the microscopic surface topography that can measure the kurtosis of the height distribution, and it is a dimensionless parameter. The $S_{ku}$ parameter involves microtopographic peaks and pits that can assess the stability of worn surfaces. As shown in Figure 18, it is a graph showing

the relationship between the surface kurtosis $S_{ku}$ and the change in the external load and the rotational speed.

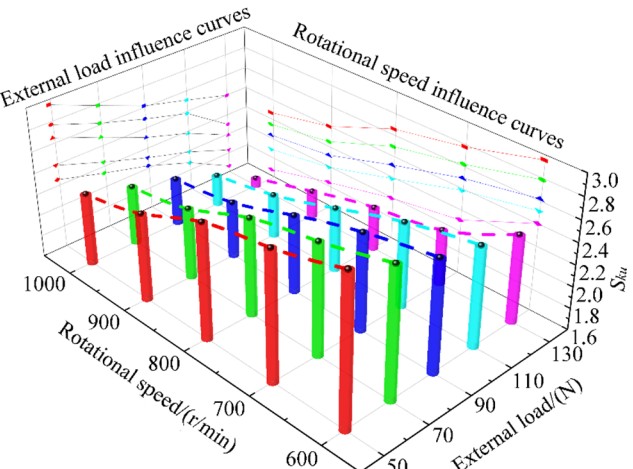

**Figure 18.** Effect of friction parameters on surface kurtosis. (red is the value of $S_{ku}$ at different rotational speeds at 50N, green is the value of $S_{ku}$ at different rotational speeds at 70N, blue is the value of $S_{ku}$ at different rotational speeds at 90N, light blue is the value of $S_{ku}$ at different rotational speeds at110N, and pink is the value of $S_{ku}$ at different rotational speeds at 130N).

The surface kurtosis $S_{ku}$ decreases with increasing external load. When the external load is low (50 N~90 N), the wear depth of the rough surface is small, and the number of peaks on the surface is large. Furthermore, the width is narrow, the distribution is wide, the sharpness of peaks and pits is large, and the average reduction in $S_{ku}$ is 0.146. When the external load is high (90 N to 110 N), the wear depth on the rough surface is large. The number of surface peaks is reduced due to wear, the peak is plastically deformed, and the width of the peak becomes wider, so the average reduction in $S_{ku}$ value becomes greater, that is 0.155. In the process of increasing the rotational speed, the intensification of extrusion and plastic deformation makes the peak prone to shear failure. The peak is plastically deformed and the wear degree is large. Therefore, the sharpness of peaks and pits gradually decreases, and the average reduction in $S_{ku}$ is 0.163.

### 4.3. Effect of Friction Parameters on the Height of the Surface Topology

Considering the change in the bearing area curves of the worn rough surface, the wear laws of the friction pair surface under different friction conditions are compared and analyzed, as shown in Figure 19. When the external load is constant, the bearing area curves decrease as the rotational speed increases. It shows that the increase in the rotational speed increases the friction contact area per unit time. During the same friction time, the same position wears more times, which reduces the surface topography height. Taking Figure 19a as an example, in the process of increasing the rotational speed from 600 r/min to 1000 r/min, due to the small proportion of the peak region and the pit region, the decrease in the peak value is not obvious, while the peak in the middle region decreases with an amplitude of 6.22 μm, the proportion of peak values in the peak region and pit region also decreases, and the proportion of peak values in the middle region increases. It shows that the number of convex peaks and deeper pits on the surface is reduced, the overall surface tends to be smooth, and the real contact area is increased, which improves the surface support performance. When the rotational speed is constant, the distribution trend of the bearing area curves after wear is similar in the process of increasing the external load from 50 N to 130 N, and there is no obvious change.

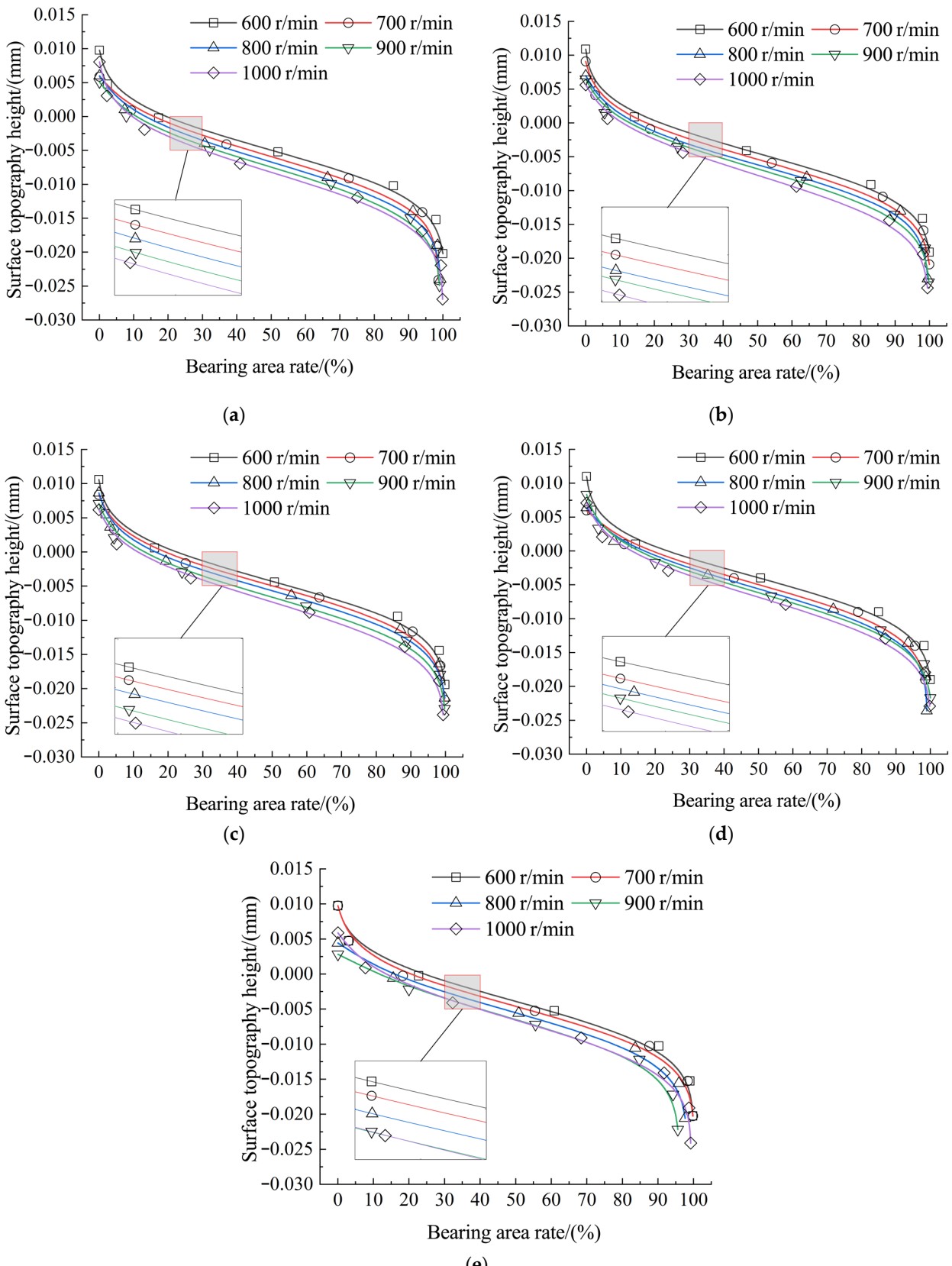

**Figure 19.** Effect of friction parameters on bearing area curves of surface topography. (**a**) The external load is 50 N; (**b**) the external load is 70 N; (**c**) the external load is 90 N; (**d**) the external load is 110 N; (**e**) the external load is 130 N.

## 5. Conclusions

In the face of the surface wear phenomenon of the friction pair in the braking process of traffic transportation, the rough surface of the friction pair is taken as the research object to analyze the change in the rough surface morphology and the wear law during the friction process. This paper comprehensively considers the different friction contact forms of rough peaks and pits in the rough surface wear process and establishes the fractal wear calculation model of the friction contact state and friction non-contact state, respectively. Image digital processing technology is adopted to complete the transformation of 3-D information from the 3-D rough surface to the 2-D plane. In addition, according to the wear calculation model, the time-varying wear calculation method of rough surfaces is proposed, the iterative calculation of the wear cycle is realized, the numerical simulation results of the surface topography of the friction pair surface during the wear process are obtained, and the influence of the friction parameters on the surface topography characteristics is analyzed. The main conclusions are as follows under the friction conditions defined in this paper.

(1) The overall peak height of the worn rough surface shows a decreasing trend, the peak value in the peak region decreases by 11.87%, the peak value in the pit region increases by 13.81%, and the peak value in the middle region decreases by 1.94%. The real contact area of the rough surface is increased, and the surface-bearing capacity is improved.

(2) The change in the wear depth of the surface topography of the friction pair before and after wear is compared and analyzed through the pin-on-disc wear experiment. The maximum surface profile height decreases significantly during the wear process, and the error of the average height value of the surface topography is 1.83%, which verifies the numerical simulation of wear effectiveness.

(3) When the rotational speed increases from 600 r/min to 1000 r/min, the average increase in wear depth is 49.15%, the surface height deviation $S_a$ increases by 3.41%, the root-mean-square deviation $S_q$ of the surface height distribution increases by 17.47%, and the decreased value of the surface kurtosis $S_{ku}$ is 0.163. When the external load increases from 50 N to 130 N, the average increase in wear depth is 43.95%, the surface height deviation $S_a$ increases by 17.82%, the surface height deviation $S_q$ increases by 5.06%, and the surface kurtosis $S_{ku}$ decreases by 0.146.

(4) The friction parameters have different effects on the surface-bearing area curves of the worn rough surface. When the external load is constant, the surface-bearing area curves decrease as the rotational speed increases. When the rotation speed is constant, the distribution trend of the surface bearing area curves after wear is similar, and there is no obvious change.

**Author Contributions:** Conceptualization, Q.H. and Z.S.; methodology, Q.H. and Z.S.; software, Q.H.; validation, Q.H. and J.Y.; formal analysis, Q.H. and J.Y.; investigation, Q.H. and L.J.; resources, Q.H. and S.Z.; data curation, Q.H.; writing—original draft preparation, Q.H. and Y.L.; writing—review and editing, Q.H. and Z.S.; visualization, J.Y. and Z.S.; supervision, Q.H. and Z.S.; project administration, Z.S. and Y.L.; funding acquisition, Z.S. and J.Y. All authors have read and agreed to the published version of the manuscript.

**Funding:** This research was funded by the National Natural Science Foundation of China (No. 51675075), the Applied Basic Research Program of Liaoning Province (No. 2022030464-JH2/1013), the Natural Science Foundation Project Program of Liaoning Province (No. 2022-BS-260), the Scientific Research Project of the Education Department of Liaoning Province (No. LJKZ0479).

**Institutional Review Board Statement:** Not applicable.

**Informed Consent Statement:** Not applicable.

**Data Availability Statement:** Not applicable.

**Conflicts of Interest:** The authors declare no conflict of interest.

## Abbreviations

| | |
|---|---|
| $a_r$ | the real contact area, mm$^2$ |
| $a_n$ | the nominal contact area of the rough surface, mm$^2$ |
| $a_{min}$ | the minimum real contact area of a single peak, mm$^2$ |
| $a_{max}$ | the maximum real contact area of a single peak, mm$^2$ |
| $A_r$ | the total real contact area of the rough surface, mm$^2$ |
| $D$ | the 3-D fractal dimension |
| $D_s$ | the 2-D fractal dimension, $D_s = D$-1 |
| $E$ | the equivalent modulus, MPa |
| $E_1$ and $E_2$ | Young's modulus of the friction pair, MPa |
| $g(x_i,y_j)$ | the gray value of the discrete unit |
| $G$ | the scaling coefficient, mm |
| $h$ | the average wear depth of the rough surface, mm |
| $\Delta h_{wear}$ | the surface wear depth per unit time, mm |
| $\Delta h_{wear}(x_i,y_j)$ | the wear depth of discrete units, mm |
| $H[x_i,y_j,z(x_i,y_j)]$ | the discrete unit data point, mm |
| $H_{wear}[x_i,y_j,z_{wear}(x_i,y_j)]$ | the discrete unit data points of the worn surface, mm |
| $k_{we}$ | the elastic wear coefficient |
| $k_{wp}$ | the plastic wear coefficient |
| $K$ | the wear coefficient of the friction pair |
| $l$ | the diameter of the real contact area, mm |
| $L$ | the sample length, where Ls is the cut-off length, mm |
| $m$ and $n$ | frequency indices of random contours |
| $n_{max}$ | the highest frequency of the spatial frequency index |
| $M$ and $N$ | number of data points |
| $M_0$ | the number of superposed ridges used to construct the surface |
| $P(x_i,y_j)$ | the coordinate of the discrete unit |
| $R$ | the curvature radius of the peak, mm |
| $R_{max}$ | the maximum peak-to-pit distance, mm |
| $S$ | the relative sliding distance of the friction pair, mm |
| $S_a$ | the surface height deviation, μm |
| $S_q$ | the root-mean-square deviation, μm |
| $S_{ku}$ | the surface kurtosis |
| $T$ | the friction time, s |
| $v$ | the relative sliding speed, m/min |
| $V$ | the wear volume of the worn material, mm$^3$ |
| $W$ | the external load, N |
| $z(x,y)$ | the surface contours height of the rough surface, where $x$ and $y$ are surface contours geometric coordinates, mm |
| $z(x_i,y_j)$ | the height value of the discrete unit, where $x_i$ and $y_j$ are used to represent the coordinate points of the discrete unit, mm |
| $z_{max}$ | the maximum peak value of the rough surface, mm |
| $z_{min}$ | the minimum peak value of the rough surface, mm |
| $z_{wearmax}$ | the maximum peak value of the worn rough surface, mm |
| $z_1$ | the peak height that will wear, mm |
| $z_2$ | the peak height that will not wear, mm |
| $z_3$ | the peak height equal to the wear boundary, mm |
| $z_{wear1}$ | the peak height of $z_1$ after wear calculation, mm |
| $z_{wear2}$ | the peak height of $z_2$ after wear calculation, mm |
| $z_{wear3}$ | the peak height of $z_3$ after wear calculation, mm |
| $z_{wear}$ | the peak value of the worn rough surface, mm |
| $\gamma$ | a parameter that determines the density of frequencies of the surface |
| $\delta$ | the normal deformation depth, mm |
| $\delta_{wear}(x_i,y_j)$ | the deformation value of the peak, mm |
| $\delta_0$ | the initial deformation of the peak, mm |
| $\lambda$ | an experimental constant |
| $\mu$ | the friction coefficient |

| $\nu_1$ and $\nu_2$ | Poisson's ratio of the friction pair |
| $\sigma_s$ | the yield limit of the worn material, MPa |
| $\sigma_y$ | the strength limit of material deformation, MPa |
| $\varphi_{m,n}$ | the uniformly distributed value of the random phase |
| $\psi$ | the domain expansion factor |

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
