# Peer review of "Time-Varying Wear Calculation Method for Fractal Rough Surfaces of Friction Pairs"

_coatings, doi:10.3390/coatings13020270_

Round 1

Reviewer 1 Report

The authors use a combination of numerical simulation and experiment to study the change laws of the surface topography during the wear process. A time-varying wear calculation method is proposed, which includes a time-varying wear calculation model for worn surface and a compensation wear calculation model for unworn surface. The control variable method is used to analyze the influence of the rotational speed and the external load on the characteristic parameters of surface topography, which provides a new approach for solving tribological problems. There is a good agreement between the results of simulation and experiments. The results of this study can be found useful for the development and manufacture of friction pairs.

The motivation of the work and the approach adopted are well. The manuscript is publishable after the below mentioned questions/comments are addressed.

Comments

1) What is the effect of temperature on the wear process? How do your results change with temperature?

2) Please compare the accuracy of the model used in this study with other models reported in the literature for prediction of the properties. 

Author Response

We are truly appreciated for your work and other reviewers’ comments and thoughtful suggestions on our manuscript. Our manuscript, "Time-varying wear calculation method for fractal rough surfaces of friction pairs" was revised according to the reviewers' comments, and the itemized response to each reviewer’s comments is attached.
Thanks very much again for your attention to our paper.

Yours sincerely,
Zhihua Sha

Response to Reviewer Comments

Point 1: What is the effect of temperature on the wear process? How do your results change with temperature?

Response 1: Thank the reviewers for the comments, in the process of the pin-on-disc wear experiment, due to the influence of friction parameters, the temperature change of the friction pair surface is small, so this paper ignores the influence of temperature on the friction pair surface wear under this friction condition. We will also conduct in-depth research on the effect of temperature on the wear of friction pairs in the future.

For details see the revised manuscript in line 408-412 on page 13.

Point 2: Please compare the accuracy of the model used in this study with other models reported in the literature for prediction of the properties.

Response 2: Indeed, the comments made by the reviewers are of great significance. We have supplemented the comparative analysis of the literature. The accuracy of the wear calculation model is analyzed by comparing the calculation results of literature [37]. In order to study the wear law obtained from the wear model, the wear data in the literature [37] and the wear data in this paper are normalized, and the effect of friction load on the surface wear law is analyzed, as shown in Figure 15. It can be seen from the figure that the increase of friction load can increase the surface wear. Compared with the wear depth calculation curve in the literature, the trend of the wear depth curve calculated in this paper is close to the wear law obtained from the test. This is because there are many factors considered in this paper. This is because there are many factors considered in this paper.

For details see the revised manuscript in line 551-560 and Figure 15 on page 18.

Besides, the authors also revised the manuscript for other shortcomings, for details see the revised manuscript.

Reviewer 2 Report

The manuscript is focused on the area: Time-varying wear calculation method for fractal rough surfaces of friction pairs.

The manuscript has 23 pages.

Overall, the manuscript is interesting.

The manuscript has the usual structure.

In the Introduction section, it is necessary to provide more information about the area being addressed.

It is necessary to better state the motivation for the addressed area of research.

The research presented is well documented. The information is clearly presented.

However, among the weaknesses of the manuscript is the discussion of the results in the context of the current state of research in the world.

This part needs to be improved.

It is also necessary to improve the part of the conclusion.

After editing, the manuscript will be interesting for the readers of the journal.

Author Response

We are truly appreciated for your work and other reviewers’ comments and thoughtful suggestions on our manuscript. Our manuscript, "Time-varying wear calculation method for fractal rough surfaces of friction pairs" was revised according to the reviewers' comments, and the itemized response to each reviewer’s comments is attached.
Thanks very much again for your attention to our paper.

Yours sincerely,
Zhihua Sha

Response to Reviewer Comments

Point 1: In the Introduction section, it is necessary to provide more information about the area being addressed. It is necessary to better state the motivation for the addressed area of research. The research presented is well documented. The information is clearly presented. However, among the weaknesses of the manuscript is the discussion of the results in the context of the current state of research in the world. This part needs to be improved. It is also necessary to improve the part of the conclusion. After editing, the manuscript will be interesting for the readers of the journal.

Response 1: The original paper lacks the necessary explanation, and we have supplemented the content in the Introduction section. 

With the development of the economy and the improvement of science and technology, the work of transportation equipment is heavy, so it puts forward higher requirements for the safety and security of mechanical equipment operation. Disc brakes are key equipment to ensure the safe operation of transport such as rail trains and buses. When the mechanical equipment is faced with the need to execute the parking order or emergency braking conditions in the actual work, the meeting adopts the mechanical braking method [1]. Disc brakes are common brake friction pairs with tribological properties such as surface damage and material wear. This phenomenon is common in the braking process of rail trains or vehicles and other traffic transportation. The uneven stress of the friction pair caused by surface wear will lead to irregular wear, reduce the braking efficiency and seriously affect the driving safety [2]. In particular, for brake friction pairs in long-term service, surface wear is an important indicator to measure the safe operation of disc brakes.

For details see the revised manuscript in line 34-132 in the Introduction section on pages 1-3, and in line 637-640 in the Conclusions section on page 22.

Besides, the authors also revised the manuscript for other shortcomings, for details see the revised manuscript.

Reviewer 3 Report

In a reviewed manuscript, the time-varying wear calculation method for fractal rough surfaces of friction pairs was presented.

The paper is sufficiently organized; however, due to the fact that there is lot of variables, reviewer propose to add the section with the nomenclature and abbreviations. This simple modification will ensure that all the variables and shorts will be in one place, which helps the readers to understand the meaning of equations faster and it will not be necessary to analyse the entire paper, to find the meaning of the variables. Additionally, it allows us to avoid situations such as a lack of unambiguity of descriptions with equations. I suggest keeping an eye on the units - those ones should be added everywhere are necessary.

The materials and methods explanation is not clear enough. For example: Why those not others parameters were adopted? What was the temperature and humidity of the environment during the implementation of the test? Furthermore, how the parameters adopted can influence the results? How many repetitions of tribological tests were performed? Furthermore, how does the adopted number of repetitions of the tests influence (from a statistical point of view) the quality of the conclusions concerning the nature of the examined phenomenon?

You wrote on page 7 line 261 that λ is an experimentally determined parameter that is related to material hardening, lubricant, material interfacing, etc., The fundamental question is - How could such different characteristics of separate factors be combined in one parameter? This requires a thorough explanation.

The article contains limited discussions, especially since there are limitations in comparison to the results of other authors; therefore, a significant discussion regarding the underlying mechanisms controlling the observed findings should be conducted.

In addition, the authors should take into consideration the following things:

What are the particular elements presented in Fig. 6; the meaning of individual elements cannot remain in the sphere of guesswork, and it may be worth considering adding the coordinate system in some cases.

 The quality of figures should be improved, especially for some figures, the units should be added to the axes description, in other cases the scales should be added to the figures (for example, Fig. 3, 12).

- Write more about the meaning of the greyscale image shown in Figure 4.

Author Response

We are truly appreciated for your work and other reviewers’ comments and thoughtful suggestions on our manuscript. Our manuscript, "Time-varying wear calculation method for fractal rough surfaces of friction pairs" was revised according to the reviewers' comments, and the itemized response to each reviewer’s comments is attached.
Thanks very much again for your attention to our paper.

Yours sincerely,
Zhihua Sha

Response to Reviewer Comments

Point 1: The paper is sufficiently organized; however, due to the fact that there is lot of variables, reviewer propose to add the section with the nomenclature and abbreviations. This simple modification will ensure that all the variables and shorts will be in one place, which helps the readers to understand the meaning of equations faster and it will not be necessary to analyse the entire paper, to find the meaning of the variables. Additionally, it allows us to avoid situations such as a lack of unambiguity of descriptions with equations. I suggest keeping an eye on the units - those ones should be added everywhere are necessary.

Response 1: Indeed, the comments made by the reviewers are of great significance. We have added the Nomenclature, see Appendix for details.

For details see the revised manuscript in line 688-689 and in the Appendix section on pages 24-25.

Point 2: The materials and methods explanation is not clear enough. For example: Why those not others parameters were adopted? What was the temperature and humidity of the environment during the implementation of the test? Furthermore, how the parameters adopted can influence the results? How many repetitions of tribological tests were performed? Furthermore, how does the adopted number of repetitions of the tests influence (from a statistical point of view) the quality of the conclusions concerning the nature of the examined phenomenon?

Response 2: We admit it might be misunderstanding caused by our inaccurate expressions in the original paper. 

  • The research content of this paper is the wear law of rough surface of friction pair. In order to compare the simulation results with the test data, the friction parameters that can be tested by the test machine are used. This friction parameter is a common parameter used by the simulation and test.
  • During testing, the room temperature was 20 â—¦C, and the relative humidity was 25%~50%.
  • According to the influence law of friction parameters, the article will introduce the influence analysis of friction load and friction speed on the friction characteristics and wear law of rough surface in the fourth section.
  • The experiment data and simulation data are extracted, and the height of rough surface topography is sampled and analyzed for five times. Furthermore, the average height value of the surface topography is calculated, and the difference between the average height before and after wear is used as the average wear depth. The error is reduced by means of multiple sampling, and from the experimental comparison results, the error is 1.83%, which is within the allowable range. At the same time, we will continue to study the influence of friction parameters in the future.

For details see the revised manuscript in line 477-503 on page 16.

Point 3: You wrote on page 7 line 261 that λ is an experimentally determined parameter that is related to material hardening, lubricant, material interfacing, etc., The fundamental question is - How could such different characteristics of separate factors be combined in one parameter? This requires a thorough explanation.

Response 3: Indeed, our inaccurate expressions caused misunderstanding in the original paper. We have supplemented the literature on the interpretation of λ. λ is an experimentally constant [34].

For details see the revised manuscript in line 208 on page 7.

Point 4: The article contains limited discussions, especially since there are limitations in comparison to the results of other authors; therefore, a significant discussion regarding the underlying mechanisms controlling the observed findings should be conducted.

Response 4: Indeed, the original paper lacks the necessary explanation, and we have supplemented the content. 

In addition, the accuracy of the wear calculation model is analyzed by comparing the calculation results of literature [37]. In order to study the wear law obtained from the wear model, the wear data in the literature [37] and the wear data in this paper are normalized, and the effect of friction load on the surface wear law is analyzed, as shown in Figure 15. It can be seen from the figure that the increase of friction load can increase the surface wear. Compared with the wear depth calculation curve in the literature, the trend of the wear depth curve calculated in this paper is close to the wear law obtained from the test. This is because there are many factors considered in this paper.

The surface height deviation Sa is raised with the increase of the external load. And when the surface topography parameters were large, the Hertzian stress increased and caused the surface to be deformed [38].

For details see the revised manuscript in line 542-543 and line 551-560 and line 570-572 on pages 17-19.

Point 5: In addition, the authors should take into consideration the following things:

What are the particular elements presented in Fig. 6; the meaning of individual elements cannot remain in the sphere of guesswork, and it may be worth considering adding the coordinate system in some cases.

The quality of figures should be improved, especially for some figures, the units should be added to the axes description, in other cases the scales should be added to the figures (for example, Fig. 3, 12).

Write more about the meaning of the greyscale image shown in Figure 4.

Response 5: We admit it might be misunderstanding caused by our inaccurate expressions in the original paper. 

We have supplemented the variables in Figure 6 and improved the expression of Figure 6.

We have improved the quality of figures (Fig. 3, 4, 12).

We have added information about the role of grayscale image in Figure 4, as follows.

The fractal dimension D of the 3-D space has a certain relationship with the fractal dimension Ds of the 2-D plane, that is, D=Ds+1. Therefore, the information on the 3-D morphology can be stored in the 2-D plane, and the 3-D fractal dimension can be calculated by measuring the 2-D fractal dimension of the 2-D plane. Based on image recognition technology, the fractal dimension of the rough surface is calculated, and the 3-D coordinate data is converted into image color data. The 3-D data point set H[xi,yj,z(xi,yj)] of the rough surface is represented by the 2-D data point set group P(xi,yj) and g(xi,yj), where P(xi,yj) is the coordinate of the discrete unit, g(xi,yj) is the gray value of the discrete unit, and g(xi,yj)∈(0,255). Therefore, the 3-D space information can be represented by a 2-D plane grayscale image, as shown in Figure 4. Furthermore, the change in the rough surface morphology during the wear process can be expressed by the grayscale change of the grayscale image. In addition, the grayscale image not only contains the plane position information, but also can store the color grayscale data. Therefore, the use of gray scale image is convenient to measure the fractal dimension of two-dimensional surface with MATLAB, and then express the fractal dimension of rough surface with three-dimensional information.

For details see the revised manuscript in line 303-318 on pages 8-9 and Figure 3, 4, 6, 12.

Besides, the authors also revised the manuscript for other shortcomings, for details see the revised manuscript.

Round 2

Reviewer 3 Report

The authors should add description of axis as well as the units to the figures presenting the surface texture (like it is realised in the case of Fig. 9.). Additionally, please add the scale to Figure 12 representing the experiment results.

Author Response

We are truly appreciated for your work and other reviewers’ comments and thoughtful suggestions on our manuscript. Our manuscript, "Time-varying wear calculation method for fractal rough surfaces of friction pairs" was revised according to the reviewers' comments, and the itemized response to each reviewer’s comments is attached.
Thanks very much again for your attention to our paper.

Yours sincerely,
Zhihua Sha

Response to Reviewer Comments

Point 1: The authors should add description of axis as well as the units to the figures presenting the surface texture (like it is realised in the case of Fig. 9.). Additionally, please add the scale to Figure 12 representing the experiment results.

Response 1: Indeed, the comments made by the reviewers are of great significance. We have added description of axis as well as the units to the figures presenting the surface textures, as well as the scale to Figure 12 representing the experiment results.

For details see the revised manuscript on Figure 12.

Besides, the authors also revised the manuscript for other shortcomings, for details see the revised manuscript.
